# Optimizing Active Tumor Targeting Biocompatible Polymers for Efficient Systemic Delivery of Adenovirus

**DOI:** 10.3390/cells10081896

**Published:** 2021-07-26

**Authors:** Jun Young Lee, Jin Woo Hong, Thavasyappan Thambi, A-Rum Yoon, Joung-Woo Choi, Yi Li, Quang Nam Bui, Doo Sung Lee, Chae-Ok Yun

**Affiliations:** 1Department of Bioengineering, College of Engineering, Hanyang University, 222 Wangsimni-ro, Seongdong-gu, Seoul 04763, Korea; junlian98@hanyang.ac.kr (J.Y.L.); thambi@hanyang.ac.kr (T.T.); ayoon@hanyang.ac.kr (A.-R.Y.); chois0802@gmail.com (J.-W.C.); 2GeneMedicine Co., Ltd., 222 Wangsimni-ro, Seongdong-gu, Seoul 04763, Korea; jhong803@gmail.com; 3Institute of Nano Science and Technology (INST), Hanyang University, Seoul 04763, Korea; 4Theranostic Macromolecules Research Center, School of Chemical Engineering, Sungkyunkwan University, Suwon 16419, Korea; liyi@zjxu.edu.cn (Y.L.); quangnam1719@gmail.com (Q.N.B.)

**Keywords:** PNLG, adenovirus, systemic administration, folate, cancer

## Abstract

Adenovirus (Ad) has risen to be a promising alternative to conventional cancer therapy. However, systemic delivery of Ad, which is necessary for the treatment of metastatic cancer, remains a major challenge within the field, owing to poor tumor tropism and nonspecific hepatic tropism of the virus. To address this limitation of Ad, we have synthesized two variants of folic acid (FA)-conjugated methoxy poly(ethylene glycol)-b-poly{N-[N-(2-aminoethyl)-2-aminoethyl]-L-glutamate (P_5_N_2_LG-FA and P_5_N_5_LG-FA) using 5 kDa poly(ethylene glycol) (PEG) with a different level of protonation (N_2_ < N_5_ in terms of charge), along with a P_5_N_5_LG control polymer without FA. Our findings demonstrate that P_5_N_5_LG, P_5_N_2_LG-FA, and P_5_N_5_LG-FA exert a lower level of cytotoxicity compared to 25 kDa polyethyleneimine. Furthermore, green fluorescent protein (GFP)-expressing Ad complexed with P_5_N_2_LG-FA and P_5_N_5_LG-FA (Ad/P_5_N_2_LG-FA and Ad/P_5_N_5_LG-FA, respectively) exerted superior transduction efficiency compared to naked Ad or Ad complexed with P_5_N_5_LG (Ad/P_5_N_5_LG) in folate receptor (FR)-overexpressing cancer cells (KB and MCF7). All three nanocomplexes (Ad/P_5_N_5_LG, Ad/P_5_N_2_LG-FA, and Ad/P_5_N_5_LG-FA) internalized into cancer cells through coxsackie adenovirus receptor-independent endocytic mechanism and the cell uptake was more efficient than naked Ad. Importantly, the cell uptake of the two FA functionalized nanocomplexes (Ad/P_5_N_2_LG-FA and Ad/P_5_N_5_LG-FA) was dependent on the complementary interaction of FA–FR. Systemically administered Ad/P_5_N_5_LG, Ad/P_5_N_2_LG-FA, and Ad/P_5_N_5_LG-FA showed exponentially higher retainment of the virus in blood circulation up to 24 h post-administration compared with naked Ad. Both tumor-targeted nanocomplexes (Ad/P_5_N_2_LG-FA and Ad/P_5_N_5_LG-FA) showed significantly higher intratumoral accumulation than naked Ad or Ad/P_5_N_5_LG via systemic administration. Both tumor-targeted nanocomplexes accumulated at a lower level in liver tissues compared to naked Ad. Notably, the nonspecific accumulation of Ad/P_5_N_2_LG-FA was significantly lower than Ad/P_5_N_5_LG-FA in several normal organs, while exhibiting a significantly higher intratumoral accumulation level, showing that careful optimization of polyplex surface charge is critical to successful tumor-targeted systemic delivery of Ad nanocomplexes.

## 1. Introduction

In the past two decades, gene therapy has been a focus of various therapeutic approaches for cancer treatment [1,2,3]. In particular, engineered oncolytic adenovirus (Ad) is one of the most promising vectors in cancer gene therapy because of its advantageous characteristics including cancer-specific replication, potent cancer cell killing effect, and high expression level of therapeutic transgenes. Oncolytic Ad can elicit potent antitumor effects when administered locally to the tumor in a similar manner to other clinically approved oncolytic viruses. Although promising as a local therapeutic agent, a complex disease like cancer in the clinical environment requires therapeutics to be systemically administrable, as many clinical tumors (both primary and distant metastases) are inaccessible or not easily accessible via the needle. Thus, a system that is capable of efficiently delivering oncolytic viruses via the systemic route to tumor tissues is necessary to further improve the clinical efficacy of oncolytic vectors. However, systemic delivery of oncolytic Ad is a challenging task due to the immunogenicity and hepatic tropism of Ad. The immunogenic viral capsid can be easily recognized by the host immune system and lead to rapid clearance from the blood as well as adverse inflammatory response, leading to poor safety and therapeutic efficacy of systemically administered oncolytic viruses. Similarly, nonspecific hepatic sequestration and accumulation of oncolytic Ad can lead to hepatotoxicity [4].

To address these safety concerns and the poor tumor-targeting ability of systemically administered Ad, many polymeric carriers, such as poly(ethylene) glycol (PEG) and biodegradable cationic polymers, have been extensively researched to mask Ad surface to reduce the immunogenicity, prevent non-specific liver uptake, prolong blood circulation time, and improve the intratumoral accumulation of systemically administered Ad [5]. Although the enhanced permeability and retention (EPR) effect mediated by the interaction of the nanomaterial coating on the Ad surface and leaky tumor vasculatures can improve the intratumoral accumulation of systemically administered Ad, this effect alone cannot optimally redirect a sufficient fraction of administered Ad to tumor tissues. Thus, the EPR-based delivery of drugs requires large doses to achieve sufficient therapeutic index and a large fraction of administered drugs will nonspecifically accumulate in healthy organs, such as liver, kidney, and lungs, ultimately causing undesirable off-target toxicity [6].

To this end, the nanomaterial-based systemic delivery of oncolytic Ad can be further improved by utilizing active targeting moieties that are complementary to factors overexpressed in the tumor microenvironment or tumor tissues [7,8,9,10,11,12]. In support of this, our group recently reported several examples where oncolytic Ad nanocomplexes with similar physiochemical properties can suffer greatly in the number of virions accumulating in tumor tissues when the active tumor-targeting moiety is absent from the surface of nanomaterials [13,14,15].

There are several other variables, such as physicochemical attributes, that must be finely tuned and optimized to decrease the probability of fast blood clearance and nonspecific distribution of nanohybrid vectors. Earlier findings demonstrate that the optimal Ad-nanocomplex size must be less than 200 nm in diameter to efficiently internalize into cells and prolong the blood circulation time of the vector by avoiding nonspecific clearance through the reticuloendothelial system (RES) [16,17,18]. Additionally, a cationic Ad nanocomplex shows higher cellular uptake and transgene expression than anionic or neutrally charged nanocomplex with similar nanomaterial composition by improving the complex attachment and internalization through the negatively charged cellular membrane [5,19,20,21,22,23,24]. The net positive charge given to therapeutic cargo by complexation with various cationic nanomaterials can also promote endosome escaping of the cargo through the proton sponge effect [5,25,26,27]. Nevertheless, several cationic polymers with excessive charges, such as high molecular weight polyetherimide (PEI) and polyamidoamine (PAMAM) without significant chemical modification or masking of amine groups, are poor for targeted systemic delivery due to their nonspecific and high affinity toward all cellular membranes, regardless of their cell types, and negatively charged serum proteins, thus leading to nonspecific cellular uptake or rapid clearance from the blood [28,29].

Based on these backgrounds, we hypothesized that a finely tuned cationic nanohybrid vector with tumor-targeting moiety on the surface may retain systemic tumor targeting ability, efficient cellular penetration ability, and improve overall transgene expression. Here, we have utilized folic acid (FA) as targeting moiety, because the folate receptor (FR) is known to be overexpressed in various cancers such as ovarian, lung, and cervical cancers [30]. Further, FA as targeting moiety has several advantages such as small ligand size, lower immunogenicity, high affinity to target tumor, and high receptor/ligand-mediated cellular internalization [31]. As the relationship between the surface charge and tumor-targeting ability of cationic nanomaterial-based delivery systems remains unclear, we have synthesized three variants of methoxy poly(ethylene glycol)-b-poly{N-[N-(2-aminoethyl)-2-aminoethyl]-L-glutamate (PNLG)-based biodegradable polymers conjugated with 5 kDa PEG and two different numbers of amine groups (2 or 5 amines) with or without folic acid (FA) as tumor-targeting moiety (P_5_N_5_LG, P_5_N_2_LG-FA, and P_5_N_5_LG-FA, respectively) to evaluate the relationship between the net cationic charge and tumor-targeting efficiency of cationic and tumor-targeted nanomaterial for the systemic delivery of Ad. Collectively, our findings suggest that careful balancing of surface amine is required to optimize tumor-targeted delivery of Ad via cationic polymers modified with tumor-targeting moiety.

## 2. Materials and Methods

### 2.1. Cancer Cells and Ad Preparations

Human cancer cell lines (A549, MCF7, and KB) and Ad E1A-expressing HEK293 cells were purchased from the American Type Culture Collection (ATCC, Manassas, VA, USA) and cultured in high glucose Dulbecco’s Modified Eagle’s Medium (DMEM; Pan biotech, Aidenbach, Germany) containing 10% fetal bovine serum (FBS; Alpha bioregen, Boston, MA, USA) at 37 °C in an incubator with 5% CO_2_. The construction and production of a replication-incompetent Ad expressing green fluorescent protein (GFP) (dE1/GFP; referred here as Ad) has been described in our previous report [32].

### 2.2. Chemicals

Methoxy poly (ethylene glycol)-amine (mPEG-NH_2)_ (M_n_ = 5000 g/mol) and NH_2_-PEG-NH_2_ (M_n_ = 5000 g/mol) were purchased from Lasan Bio Inc., (Arab, AL, USA). L-glutamic acid γ-benzyl ester (BLG), 2-hydroxypyridine (2-HP), pyridine, *N*,*N*′-dicyclohexylcarbodiimide (DCC), folic acid (FA), chloroform, dimethyl sulfoxide (DMSO), *N*,*N*′-dimethylformamide (DMF), and tetrahydrofuran (THF) were purchased from Sigma Aldrich (St. Louis, MO, USA). Triphosgene, diethylenetriamine (DETA), and pentaethylenehexamine (PEHA) were obtained from Tokyo Chemical Industry (Tokyo, Japan).

### 2.3. Synthesis of BLG-NCA

The γ-benzyl-L-glutama-N-carboxy (BLG-NCA) monomer was prepared using the Fuchs–Farthing method. Briefly, BLG (5 g, 21 mmol) was suspended in anhydrous THF (50 mL) and slowly heated to 50 °C under nitrogen atmosphere. Thereafter, triphosgene (2.5 g, 8.4 mmol) was added slowly and stirred for 3 h to get a clear solution. The crude product was precipitated in hexane (500 mL) and stored at −20 °C overnight. The precipitate was filtered, washed with excess hexane, and dried under vacuum; the obtained BLG-NCA was stored at −20 °C until use.

### 2.4. Synthesis of PNLG

The PNLG polymer was synthesized using a two-step process. In the first step, poly(ethylene-glycol)-poly(γ-benzyl-L-glutamate) (PEG-PBLG) was prepared by ring-opening polymerization. In the second step, PNLG was obtained by aminolysis of mPEG-PBLG diblock copolymers with DETA or HEPA.

The mPEG-PBLG diblock copolymer was synthesized by the ring-opening polymerization of BLG-NCA in the presence of mPEG-NH_2_. In brief, mPEG-NH_2_ (0.2 g, 0.04 mmol) and BLG-NCA (0.42 g, 1.6 mmol) were reacted in CHCl_3_ at room temperature for 72 h under nitrogen atmosphere. The reaction mixture was precipitated in excess cold ether, filtered, and dried under vacuum to obtain mPEG-PBLG deblock copolymer.

The cationic copolymer was obtained by aminolysis of mPEG-PBLG diblock copolymer in the presence of DETA or PEHA. To prepare PNLG copolymer, mPEG-PBLG (1 mmol) and 2-HP (5 mmol) were dissolved in DMF and stirred at room temperature. Subsequently, DETA or PEHA was injected via syringe and the reaction mixture was heated to 50 °C, and the aminolysis was continued for 72 h. Thereafter, the yellowish viscous oil was cooled and transferred to a dialysis membrane tube (MWCO: 8000 Da) and dialyzed against 0.05 M HCl (1 day, 3 changes) followed by deionized water (2 days, 6 changes). The final product was passed through a 0.45 µm filter and lyophilized to obtain PNLG copolymer (denoted as P_5_N_5_LG).

### 2.5. Synthesis of PNLG-FA

The synthesis of PNLG-FA (denoted as P_5_N_5_LG-FA and P_5_N_2_LG-FA) was identical to that of PNLG except for the FA conjugation. FA was conjugated to one end of NH_2_-PEG-NH_2_. In short, FA (35 mg, 0.08 mmol) was dissolved in anhydrous DMSO in the presence of pyridine (10 µL); DCC (18 mg, 0.08 mmol) in DMSO was slowly added and stirring continued at RT for 15 min. Then, NH_2_-PEG-NH_2_ (400 mg, 0.08 mmol) in DMSO was slowly added and continued the reaction 24 h in dark condition. The insoluble dicyclohexylurea byproduct was removed by filtration, and the filtrates were transferred to a dialysis membrane tube (MWCO: 3500 Da) and dialyzed against excess water for 2 days (8 changes). Finally, the dialysate was passed through a 0.45-µm filter and lyophilized to obtain FA-PEG-NH_2_.

To obtain PNLG-FA, FA-PEG-NH_2_ was used as the macroinitiator, and synthesis was carried out and identical to the PNLG preparation.

### 2.6. Cytotoxicity of Polymers

KB, MCF7, and A549 cells were seeded at 50~60% confluence in 24-well plates, then incubated with 25 kDa PEI (50 μg/mL) or one of the three PNLG-based polymers (P_5_N_5_LG, P_5_N_2_LG-FA, or P_5_N_5_LG-FA) at various concentrations (1~50 μg/mL). At 2 days following polymer treatment, 200 μL of 2 mg/mL 3-(4,5-dimethylthiazol-2-yl)-2,5-diphenyl tetrazolium bromide (MTT) was added to each well and incubated for 4 h at 37 °C. The precipitate was dissolved in 500 μL DMSO. Plates were read on a microplate reader (Tecan Infinite M200; Tecan Deutschland GmbH, Crailsheim, Germany) at 540 nm.

### 2.7. Transduction Efficiency of Ad and Ad Nanocomplexes

The FR-overexpressing cancer cells (KB and MCF7), FR-negative cancer cells (A549), were seeded on 24-well plates at 80% confluence. Ad nanocomplexes (Ad/P_5_N_5_LG, Ad/P_5_N_2_LG-FA, or Ad/P_5_N_5_LG-FA) were formed with Ad:polymer molar ratio of 1 × 10^5^, 5 × 10^5^, 1 × 10^6^, 5 × 10^6^ through electrostatic interaction by carefully mixing the Ad (A549-50 MOI, KB-20 MOI, and MCF7-200 MOI) with one of the three PNLG-based polymers (P_5_N_5_LG, P_5_N_2_LG-FA, and P_5_N_5_LG-FA) and incubated at room temperature for 30 min. The cells were then treated with the naked Ad or Ad nanocomplexes (Ad/P_5_N_5_LG, Ad/P_5_N_2_LG-FA, or Ad/P_5_N_5_LG-FA) in a serum-free condition for 4 h at 37 °C. Subsequently, the media were removed and exchanged with fresh media containing 5% FBS. The treated cells were further incubated for 2 days, then photographed under a fluorescence microscope (Olympus IX81; Olympus Optical, Tokyo, Japan).

### 2.8. Physiochemical Properties of Ad and Ad Nanocomplexes

The average particle sizes and surface charges of naked Ad or Ad nanocomplexes (Ad/P_5_N_5_LG, Ad/P_5_N_2_LG-FA, or Ad/P_5_N_5_LG-FA formed with Ad:polymer molar ratio of 5 × 10^5^) were determined by the Zetasizer 3000HS (Malvern Instrument Inc., Worcestershire, UK) with a He-Ne Laser beam (633 nm, fixed scattering angle of 90°) at room temperature. The surface charges of the nanomaterials (P_5_N_5_LG, P_5_N_2_LG-FA, and P_5_N_5_LG-FA) were measured by Zetasizer 3000HS as described above. The obtained sizes and surface charges are the average values of 3 runs for independently prepared triplicate of each sample.

### 2.9. Competition Assay of Ad and Ad Nanocomplexes

KB cells were plated at 1.5 × 10^5^ cells per well in a 12-well plate. At 24 h after cell seeding, the culture media were replaced with serum-free DMEM, then the cells were pretreated with PBS, free folate (5 and 20 µg/mL), or CAR-specific antibody (Ab; 20 and 50 µg/mL) at 4 °C for 1 h. Naked Ad or Ad nanocomplexes (Ad/P_5_N_5_LG, Ad/P_5_N_2_LG-FA, or Ad/P_5_N_5_LG-FA) were added to the media (50 in FA-treated groups and 100 MOI for CAR Ab-treated groups) and incubated at 37 °C for 1 h. The media were removed by vacuum suction, the cells were washed three times with PBS, and then replaced with DMEM containing 5% FBS. At 24 h after the Ad or Ad nanocomplex treatment, the cells were observed by fluorescence microscope. GFP expression levels were quantified by FACS Calibur analyzer (BD Bioscience, San Jose, CA, USA) using the CellQuest software (BD Bioscience). Data from 10,000 cell events were collected and analyzed for each sample.

### 2.10. Cellular Uptake of Ad and Ad Nanocomplexes

Ad (200 MOI) was labeled with fluorescein isothiocyanate (FITC; Sigma-Aldrich) for 4 h at room temperature, then dialyzed (10 kDa MWCO, Slide-A-Lyzer™ Dialysis Cassettes, Life Technologies) overnight in an ice-cold PBS to remove unreacted FITC. KB and A549 cells were treated with Ad-FITC, Ad-FITC/P_5_N_5_LG, Ad-FITC/P_5_N_2_LG-FA, or Ad-FITC/P_5_N_5_LG-FA formed with Ad:polymer molar ratio of 5 × 10^5^ for 2 h, then washed 3 times with PBS to remove any reagent bound to the outer cell membrane. FITC levels were quantified by flow cytometry, as described above.

### 2.11. Neutralizing Antibody Assay

To assess whether Ad nanocomplexes can be protected from neutralization by anti-Ad antibodies, neutralization of GFP-expressing Ad was measured in vitro. After Ad-neutralizing antibody was diluted in PBS (1/100, 1/200, 1/500), diluted antibody was then mixed with naked Ad or Ad nanocomplexes (Ad/P_5_N_5_LG, Ad/P_5_N_2_LG-FA, or Ad/P_5_N_5_LG-FA) for 1 h at 37 °C. KB cells were treated with Ad or Ad nanocomplexes pre-treated with Ad-specific neutralizing Ab (100 MOI) for 48 h at 37 °C. The degree of GFP expression was then observed and analyzed by the Incucyte Zoom live cell analysis system (Essen Bio-Science, Ann Arbor, MI, USA).

### 2.12. Pharmacokinetics of Ad and Ad Nanocomplexes

To assess the rate of Ad clearance from the blood of mice, real-time quantitative PCR (Q-PCR) was performed on whole blood samples from mice that were intravenously injected once with 1 × 10^10^ VP of Ad or Ad nanocomplexes (*n* = 3 per group), as previously described [33,34]. In brief, 100 μL of whole blood were collected from the retro-orbital plexus of nude mice at 5 min, 10 min, 20 min, 30 min, 1 h, 6 h, or 24 h after the systemic injection. Total DNA from an aliquot of whole blood was extracted using the QIAamp DNA Blood Mini Kit (Qiagen, Hilden, Germany). The number of Ad genomes was quantitated using Applied Biosystems 7500 (Applied Biosystems, Foster City, CA, USA).

### 2.13. Biodistribution of Ad and Ad Nanocomplexes

To evaluate the antitumor efficacy of different treatments, the KB human xenograft tumor model was established subcutaneously by injecting KB cells (5 × 10^6^ cells) under the abdominal skin of 6- to 8-week-old male athymic nude mice (Orientbio, Seong-nam, Korea). Once the average tumor volumes reached 200 mm^3^, mice were randomized into five groups (PBS, Ad, Ad/P_5_N_5_LG, Ad/P_5_N_2_LG-FA, or Ad/P_5_N_5_LG-FA). Tumor-bearing mice were intravenously injected a total of three times with 1 × 10^10^ VP of naked Ad, Ad/P_5_N_5_LG, Ad/P_5_N_2_LG-FA, or Ad/P_5_N_5_LG-FA every other day (*n* = 3). The lung, heart, kidney, brain, muscle, spleen, liver, and tumor tissues were harvested 24 h after the third injection, then the DNA was extracted from the tissues using the QIAamp DNA Mini Kit (Qiagen) according to the manufacturer’s instructions [35]. The number of viral genomes in each sample was assessed by Q-PCR, as described above.

### 2.14. Immunohistochemical Analysis

Tumor and normal tissues (heart, kidney, liver, spleen) were collected from KB human xenograft tumor model at 2 days after the last treatment, embedded in paraffin, and sectioned at 4-µm thickness for immunohistochemical staining. All tissue sections were incubated with a goat-anti-adenovirus hexon polyclonal primary Ab (Chemicon, USA). After washing, the sections were incubated with anti-goat IgG-HRP Ab (6160-05; Southern Biotech) as a secondary Ab, and then counterstained with Meyer’s hematoxylin (Sigma). The slides were then examined under a fluorescence microscope (IN Cell analyzer 2200).

### 2.15. Statistical Analysis

The data was expressed as the mean ± standard deviation (SD) where indicated. Comparisons between two samples were analyzed for homogeneity of variance using Levene’s test and analyzed by student’s *t*-test. Groups with *p* values less than 0.05 were considered statistically significant.

## 3. Results

### 3.1. Synthesis of Biocompatible Polymers

The biocompatible and biodegradable PNLG copolymers were synthesized using a two-step process as shown in Figure 1A [36]. PNLG-FA copolymers comprised of FA were synthesized by the controlled modification of bifunctional NH_2_-PEG-NH_2_, and subsequent two-step process yields PNLG-FA (Figure 1B). To synthesize FR-targeted PNLG, FA was first reacted with NH_2_-PEG-NH_2_ through DCC coupling chemistry. The mono-functionalization of NH_2_-PEG-NH_2_ by FA was confirmed by ^1^H NMR spectrum, which showed ~95% of FA conjugation (Figure 1C). The FA-PEG-NH_2_ polymer was used as a macroinitiator to afford the diblock copolymer, FA-PEG-PBLG, through ring-opening polymerization of BLG-NCA. ^1^H NMR spectrum in Figure 1D shows the presence of FA, PEG, and PBLG characteristics peaks, indicating successful polymer synthesis. The degree of polymerization (DP) of PBLG block was calculated by comparing the integral values of PEG at 3.60 ppm with the benzylic methylene protons of PBLG at 5.02 ppm. Regardless of PEG molecular weight, the DP of PBLG was found to be ~40, indicating the successful preparation of uniform diblock copolymers. The final hydrophilic and cationic copolymers (PNLG-FA) were obtained by aminolysis reaction of FA-PEG-PBLG. To obtain PNLG-FA copolymers, aminolysis of FA-PEG-PBLG was done using DETA or HEPA in the presence of 2-HP. The ^1^H NMR spectrum in Figure 1E,F shows the appearance of new characteristics peaks at 2.55 ppm corresponding to the peaks of DETA or HETA. More importantly, the complete disappearance of aromatic and benzylic methylene protons was observed, which indicated the complete consumption of ester groups by amines. The molecular weights of all the polymers were calculated based on the ^1^H NMR results, and these data are listed in Table 1.

### 3.2. In Vitro Cytotoxicity Profile of P_5_N_5_LG, P_5_N_2_LG-FA, and P_5_N_5_LG-FA

One of the largest concerns regarding cationic nanomaterials is the potential risk for nonspecific cytotoxicity. In particular, a higher cationic charge of nanomaterial has frequently been associated with a higher level of cytotoxicity [37]. Thus, we have compared the toxicity profile of P_5_N_5_LG, P_5_N_2_LG-FA, and P_5_N_5_LG-FA with potentially different net cationic surface charges (Figure 2A) due to the difference in the number of surface amine groups and presence/absence of targeting moiety on the nanomaterial. As shown in Figure 2B, significant toxicity was not observed in all cells treated with P_5_N_2_LG-FA, or P_5_N_5_LG-FA compared to P_5_N_5_LG up to 50 µg/mL. This was in stark contrast to substantial cytotoxicity mediated by 25 kDa PEI (>90% cytotoxicity in all tested cell lines) at a polymer concentration of 20 µg/mL and higher (*p* < 0.001). These results suggest that the difference in numbers of surface amines and the presence of targeting moiety did not affect the cytotoxicity of PNLG.

### 3.3. Transduction Efficacy of Ad Complexed with P_5_N_5_LG, P_5_N_2_LG-FA, and P_5_N_5_LG-FA

To optimize the overall FR targeting and transduction efficacy of Ad nanocomplexes, FR-positive (KB and MCF) and -negative (A549) cancer cells were transduced with naked Ad or Ad nanocomplexes (Ad/P_5_N_5_LG, Ad/P_5_N_2_LG-FA, or Ad/P_5_N_5_LG-FA formed with Ad:polymer molar ratio ranging from 1 × 10^5^ to 5 × 10^6^). As shown in Figure 3, Ad nanocomplexes (Ad/P_5_N_5_LG, Ad/P_5_N_2_LG-FA, and Ad/P_5_N_5_LG-FA) induced a higher level of GFP expression than naked Ad across most of the Ad:polymer molar ratios in cancer cell lines.

More importantly, both FR-targeted Ad/P_5_N_2_LG-FA and Ad/P_5_N_5_LG-FA showed a higher level of transduction than Ad/P_5_N_5_LG in FR-positive KB and MCF7 cells while showing a markedly lower level of transduction than Ad/P_5_N_5_LG in FR-negative A549 cells at Ad:polymer molar ratio of 5 × 10^5^. These results suggest that transduction of Ad mediated by cationic nanomaterial can be guided toward target cancer cells by engrafting FA as a targeting moiety on the outer shell of the polymeric layer. As the highest transduction efficacy of targeted Ad nanocomplexes (Ad/P_5_N_2_LG-FA and Ad/P_5_N_5_LG-FA) was achieved in FR-positive cancer cells at Ad:polymer ratio of 5 × 10^5^, this polymer ratio was chosen as the optimal ratio for all nanocomplexes in subsequent experiments.

### 3.4. Physiochemical Properties of Ad Complexed with P_5_N_5_LG, P_5_N_2_LG-FA, and P_5_N_5_LG-FA

Physiochemical attributes, such as size and surface zeta potential of the nanohybrid vector have been reported to affect systemic delivery efficacy due to these factors being a regulator of clearance by RES or intratumoral accumulation of nanoparticles in tumor tissues via the EPR effect. Thus, we have examined how the level of surface amination and presence of FA on nanomaterial surface affect the physiochemical attributes of final Ad nanocomplexes combining replication-incompetent Ad and one of the P_5_N_5_LG, P_5_N_2_LG-FA, and P_5_N_5_LG-FA nanomaterials (Ad/P_5_N_5_LG, Ad/P_5_N_2_LG-FA, and Ad/P_5_N_5_LG-FA, respectively). As shown in Figure 4, coating the surface of Ad with control cationic polymer lacking tumor targeting moiety (P_5_N_5_LG) led to a significant increase in net surface charge and size compared to those of naked Ad, suggesting that cationic nanomaterial effectively masked the Ad capsid. The diameters of Ad/P_5_N_2_LG-FA and Ad/P_5_N_5_LG-FA complexes were significantly higher than those of Ad/P_5_N_5_LG, suggesting that FA conjugation to the surface of PNLG core increases the size of the complex. In terms of surface charge of nanohybrid vectors, Ad/P_5_N_5_LG-FA showed a higher net surface charge than Ad/P_5_N_2_LG-FA, suggesting that the number of amines affects the net positive charge of the nanohybrid complex (*p* < 0.01).

### 3.5. FR Availability and Its Effect on the Internalization of Ad Complexed with P_5_N_5_LG, P_5_N_2_LG-FA, and P_5_N_5_LG-FA

To assess receptor-dependence in the internalization of each Ad nanocomplexes, CAR- and FR-positive KB cells were transduced with naked Ad or Ad nanocomplexes (Ad/P_5_N_5_LG, Ad/P_5_N_2_LG-FA, or Ad/P_5_N_5_LG-FA) in the presence or absence of either CAR-specific Ab or free FA. As shown in Figure 5A, pretreatment of KB cells with free FA did not highly affect the transduction of naked Ad or Ad/P_5_N_5_LG nanocomplex, which lacks FA to target FR expressed in KB cells, compared to Ad/P_5_N_2_LG-FA or Ad/P_5_N_5_LG-FA complexes. In marked contrast, both FR-targeted nanocomplexes Ad/P_5_N_2_LG-FA and Ad/P_5_N_5_LG-FA showed a reduction in GFP expression (*p* < 0.001). These results and those of Figure 3 strongly indicate that the transduction of FR-targeted nanocomplexes is strongly reliant on the interaction between FA on the surface of nanomaterial and FR overexpressed on cancer cells. In regard to the pretreatment of KB cells with CAR-specific Ab, only naked Ad showed a significant reduction of GFP expression level in a Ab concentration-dependent manner, whereas there was no observable decrease in GFP expression level by FA-targeted Ad nanocomplexes (Ad/P_5_N_2_LG-FA, or Ad/P_5_N_5_LG-FA) (Figure 5B). Interestingly, pretreatment with CAR-specific Ab also led to a minimal level of Ab concentration-dependent reduction in GFP expression level by Ad/P_5_N_5_LG treatment (albeit at a much lower level of reduction in GFP expression than those of naked Ad). These results suggest that, at this Ad:polymer molar ratio, there may be Ad particles that are not fully coated by nanomaterial when FA is not incorporated into nanomaterial formulation, thus leading to minor CAR-dependence in transduction by Ad/P_5_N_5_LG, whereas the two FA-incorporated Ad nanocomplexes were internalized into cancer cells completely independent of CAR.

To compare FR expression level-dependent cellular uptake of naked Ad and Ad nanocomplexes, the cellular uptake efficiency of fluorescein isothiocyanate (FITC)-tagged Ad (Ad-FITC) and Ad nanocomplexes (Ad-FITC/P_5_N_5_LG, Ad-FITC/P_5_N_2_LG-FA, and Ad-FITC/P_5_N_5_LG-FA) was analyzed in FR-positive KB cells and FR-negative A549 cells by flow cytometry. As shown in Figure 5C, Ad-FITC/P_5_N_2_LG-FA and Ad-FITC/P_5_N_5_LG-FA exhibited significantly higher cellular uptake than Ad-FITC/P_5_N_5_LG in FR-overexpressing KB cells (*p* < 0.001), whereas these two targeted nanocomplexes showed much lower level of internalization than those of Ad-FITC/P_5_N_5_LG in FR-negative A549 cells. These results suggest that cellular uptake efficiency of nanocomplexes composed with cationic and FR-targeted nanomaterials (P_5_N_2_LG-FA and P_5_N_5_LG-FA) on the surface of Ad is greatly affected by cellular FR expression level, and these results are in line with our findings from Figure 3 and Figure 5A. Of note, Ad-FITC/P_5_N_5_LG-FA exhibited much higher cellular uptake than either the Ad-FITC/P_5_N_2_LG or Ad-FITC/P_5_N_2_LG-FA in both FR-positive and FR-negative cancer cells (*p* < 0.001). These results suggest that P_5_N_2_LG-FA could possess better FR specificity than P_5_N_5_LG-FA due to a lower level of surface amines and nonspecific charge-mediated cellular internalization.

### 3.6. Shielding of Ad Complexed with P_5_N_5_LG, P_5_N_2_LG-FA, and P_5_N_5_LG-FA from Ad-Neutralizing Ab

We examined whether Ad nanocomplexes (Ad/P_5_N_5_LG, Ad/P_5_N_2_LG-FA, or Ad/P_5_N_5_LG-FA) could overcome rapid clearance induced by Ad-specific neutralizing Ab, which is vital for blood circulation time. As shown in Figure 6, naked Ad pre-treated with Ad-specific neutralizing Ab (1/200 dilution) lost activity, showing significantly reduced Ad-mediated GFP expression in KB cells by 90%. In contrast, Ad nanocomplexes (Ad/P_5_N_5_LG, Ad/P_5_N_2_LG-FA, or Ad/P_5_N_5_LG-FA) with the treatment of 1/200 dilution of Ad-specific Ab showed a GFP expression reduction by 59%, 52%, and 2%, respectively, demonstrating that PNLG polymers can shield Ad from Ad-specific neutralizing. Of note, Ad/P_5_N_5_LG-FA showed much better protection from Ad-specific neutralization compared with either Ad/P_5_N_5_LG or Ad/P_5_N_2_LG-FA. Since neutralizing Ab inactivates Ad and causes failure of repeated administration of Ad vectors [38], this strong reduction of neutralization by Ad-specific neutralizing Ab could increase blood retention of Ad nanocomplexes in the bloodstream through systemic administration.

### 3.7. Blood Retention and Biodistribution Profile of Ad Complexed with P_5_N_5_LG, P_5_N_2_LG-FA, and P_5_N_5_LG-FA

Most cationic nanomaterials with excessive cationic charge are poor for targeted systemic delivery due to their nonspecific affinity toward negatively charged cellular and blood components, ultimately leading to rapid clearance from the blood and poor tumor targeting [39]. Thus, to compare how different nanomaterials with a varying number of amines and presence of targeting moiety affect the blood retention time of systemically administered Ad, 1 × 10^10^ viral particles (VP) of naked Ad or Ad nanocomplexes (Ad/P_5_N_5_LG, Ad/P_5_N_2_LG-FA, or Ad/P_5_N_5_LG-FA) were systemically administered via intravenous route and the number of VP was determined from the blood at multiple time points via quantitative real-time PCR. As shown in Figure 7A, Ad nanocomplexes utilizing targeted cationic nanomaterials (Ad/P_5_N_2_LG-FA and Ad/P_5_N_5_LG-FA) showed a higher level of viral retention than nontargeted Ad/P_5_N_5_LG at all-time points. In particular, Ad/P_5_N_2_LG-FA showed a 4.0-fold higher level of viral retention at 1 h post-injection compared to Ad/P_5_N_5_LG (*p* < 0.001), indicating that targeting moiety FA may protect polymers from nonspecific binding toward negatively charged cellular and blood components. Furthermore, all Ad nanocomplexes (Ad/P_5_N_5_LG, Ad/P_5_N_2_LG-FA, and Ad/P_5_N_5_LG-FA) exhibited a significantly higher quantity of virions being retained in blood at all respective time points when compared with naked Ad. At 30 min post-injection, Ad/P_5_N_5_LG, Ad/P_5_N_2_LG-FA, and Ad/P_5_N_5_LG-FA showed 6.8 × 10^3^-, 1.4 × 10^4^-, and 8.8 × 10^3^-fold higher retention than naked Ad (*p* < 0.001). The difference in blood retention of virions between naked Ad and nanocomplexes remained up to 24 h post-injection with Ad nanocomplexes still exhibiting 1.9 × 10^3^, 2.4 × 10^3^-, and 2.4 × 10^−3^-fold higher retention than naked Ad, respectively (*p* < 0.001). These results demonstrate that PNLG-based polymer can efficiently mask the Ad surface, thus resulting in superior blood circulation time.

Subsequently, we evaluated the tumor homing ability of various nanomaterials on the Ad surface by examining the biodistribution profile of systemically administered naked Ad, Ad/P_5_N_5_LG, Ad/P_5_N_2_LG-FA, and Ad/P_5_N_5_LG-FA in mice harboring subcutaneous KB tumors. As shown in Figure 7B, the highest quantity of virions was detected in the liver for naked Ad group due to the well-established hepatic tropism of systemically administered serotype 5 Ads [14,15,40]. In contrast, all nanocomplexes (Ad/P_5_N_5_LG, Ad/P_5_N_2_LG-FA, and Ad/P_5_N_5_LG-FA) showed significantly decreased levels of viral particles (98.8%, 92.5%, and 88.3% reduction compared to naked Ad, respectively; *p* < 0.001 or *p* < 0.05) in the liver. In marked contrast, Ad/P_5_N_5_LG, Ad/P_5_N_2_LG-FA, and Ad/P_5_N_5_LG-FA showed 1.0 × 10^2^-, 4.7 × 10^4^-, and 4.1 × 10^3^-fold higher level of intratumoral accumulation than naked Ad (*p* < 0.001). More importantly, FR-targeted nanocomplexes (Ad/P_5_N_2_LG-FA, and Ad/P_5_N_5_LG-FA) showed 460- and 41-fold higher tumor accumulation than Ad/P_5_N_5_LG (*p* < 0.001). Of note, the accumulation of viral particle in normal tissue was minimal compared with tumor sites, showing 0.02% (Ad/P_5_N_2_LG-FA) and 3.64% (Ad/P_5_N_5_LG-FA) of viral particles detected in the heart tissue compared with tumor sites, respectively. A similar trend was also observed in the kidney. Further, we have observed no detectable level of Ad hexon protein in these normal tissues shown by IHC staining, whereas a high level of Ad particles was observed in tumor tissue (Figure 7C). Additionally, in Table 2, the tumor-to-liver ratio of Ad/P_5_N_2_LG-FA and Ad/P_5_N_5_LG-FA was 72.0- and 4.0-fold higher compared to Ad/P_5_N_5_LG, showing the improved tumor-specific accumulation through FA–FR specific interaction.

## 4. Discussion

To overcome the current limitations of the systemic delivery of Ad, such as nonspecific liver uptake of Ad, short circulatory half-life of virus in blood, and low accumulation at target tumor site, we generated three different Ad nanocomplexes (Ad/P_5_N_5_LG, Ad/P_5_N_2_LG-FA, or Ad/P_5_N_5_LG-FA) using PEGylated PNLG variants with or without tumor-targeting moiety, FA. All of the generated PNLG polymers (P_5_N_5_LG, P_5_N_2_LG-FA, and P_5_N_5_LG-FA) were PEGylated using a 5-kDa PEG to minimize cytotoxicity based on our previous investigation comparing Ad or plasmid DNA complexed with PNLG containing 2 kDa, 3.4 kDa, or 5 kDa PEG [38], which demonstrated that PEGylation of PNLG with higher molecular weight PEG correlated to a lower level of cytotoxicity through efficient charge shielding of the polymer. In support of that, P_5_N_5_LG, P_5_N_2_LG-FA, and P_5_N_5_LG-FA nanomaterials induced no significant cytotoxicity in several cell lines (Figure 2B), demonstrating that PEGylation can attenuate cytotoxicity of cationic nanomaterials. These findings are consistent with other reports showing that shielding of cationic functional groups by PEGylation or other surface modification strategies can improve the safety profile of nanomaterials [15,40,41,42].

Physiochemical properties, such as particle size and surface charge, have been shown to play a pivotal role in the cellular uptake of nanoparticles: in general, nanoparticles with higher surface charge have been shown to bind more strongly to the cell membrane and internalize more effectively due to a higher level of electrostatic interactions between the anionic membrane and cationic nanoparticles [43]. In line with these reports, Ad/P_5_N_5_LG-FA complex with a higher cationic charge in comparison to Ad/P_5_N_2_LG-FA induced a higher level of GFP expression in cancer cell lines at lower moral ratios (Figure 3), showing that higher level of amine and surface protonation of the complex can improve the gene transfer efficiency of the nanocomplex in vitro. Of note, both the FR-targeted nanocomplexes (Ad/P_5_N_2_LG-FA and Ad/P_5_N_5_LG-FA) induced a lower level of GFP expression level in A549 cells with low FR expression level in comparison to Ad/P_5_N_5_LG, despite the tumor-targeted nanocomplexes having a similar or higher cationic charge than Ad/P_5_N_5_LG, respectively (Figure 3 and Figure 4). This result in conjunction with FR-specific internalization of Ad/P_5_N_2_LG-FA and Ad/P_5_N_5_LG-FA complexes shown in Figure 5 demonstrate that the inclusion of tumor-targeting moieties to the surface of cationic nanomaterial can attenuate charge-dependent nonspecific uptake while simultaneously improving tumor-targeted internalization of nanocomplexes. Furthermore, Ad/P_5_N_2_LG-FA and Ad/P_5_N_5_LG-FA complexes induced a lower reduction of GFP expression in KB cells compared to naked Ad in the presence of anti-Ad Ab, proving that PNLG polymers can efficiently protect Ad from neutralization by anti-Ad Ab.

Although the Ad/P_5_N_5_LG-FA complex induced a higher level of transgene expression level than Ad/P_5_N_2_LG-FA in vitro, Ad/P_5_N_2_LG-FA was shown to exhibit several superior attributes, such as tumor-targeted systemic delivery efficacy and blood retention profile, over Ad/P_5_N_5_LG-FA in vivo (Figure 7). Firstly, intravenously administered Ad/P_5_N_2_LG-FA was retained in the bloodstream at a maximum of 1.86-fold higher level than Ad/P_5_N_5_LG-FA from 5 min to 24 h post-administration (Figure 7A), suggesting that the higher cationic charge of Ad/P_5_N_5_LG-FA may promote more rapid blood clearance of the complex. This is in agreement with reports demonstrating that highly cationic nanoparticles are rapidly cleared from blood circulation [44,45]. Secondly, systemic administration of Ad/P_5_N_5_LG-FA induced a higher level of off-target accumulation in normal organs (heart, kidney, and muscle) than Ad/P_5_N_2_LG-FA, as well as resulted in a lower level of intratumoral virion accumulation (Figure 7B,C), indicating that excess cationic charge can impede the tumor-targeted systemic delivery of Ad nanocomplex. These results are also in line with in vitro characterization where Ad/P_5_N_5_LG-FA was shown to nonspecifically internalize into A549 (low FR expressing cell line) at a higher level than Ad/P_5_N_2_LG-FA (Figure 5C) as well as in FR-overexpressing KB cells that were pre-treated with FA at 20 μg/mL (Figure 5A), demonstrating the superior tumor-targeting ability of Ad/P_5_N_2_LG-FA. These findings are in agreement with other reports where an excess cationic charge of nanomaterials correlated with poorer tumor-targeting ability [46], owing to nonspecific interactions with serum components and blood cells that nullify material function. Although Ad/P_5_N_2_LG-FA was superior to Ad/P_5_N_5_LG-FA, it should be noted that both tumor-targeted nanocomplexes (Ad/P_5_N_2_LG-FA and Ad/P_5_N_5_LG-FA) induced better intratumoral accumulation of the virus than the non-targeted complex (Ad/P_5_N_5_LG) or naked Ad, demonstrating that inclusion of active tumor-targeting moieties to nanomaterial surface can induce higher tumor-targeted delivery of therapeutic cargos in comparison to solely relying on passive tumor targeting by EPR effect. Collectively, these findings demonstrate that tumor-targeted systemic delivery of Ad using nanomaterial requires careful optimization of the net surface charge and inclusion of tumor-targeting moieties.

## Figures and Tables

**Figure 1 cells-10-01896-f001:**
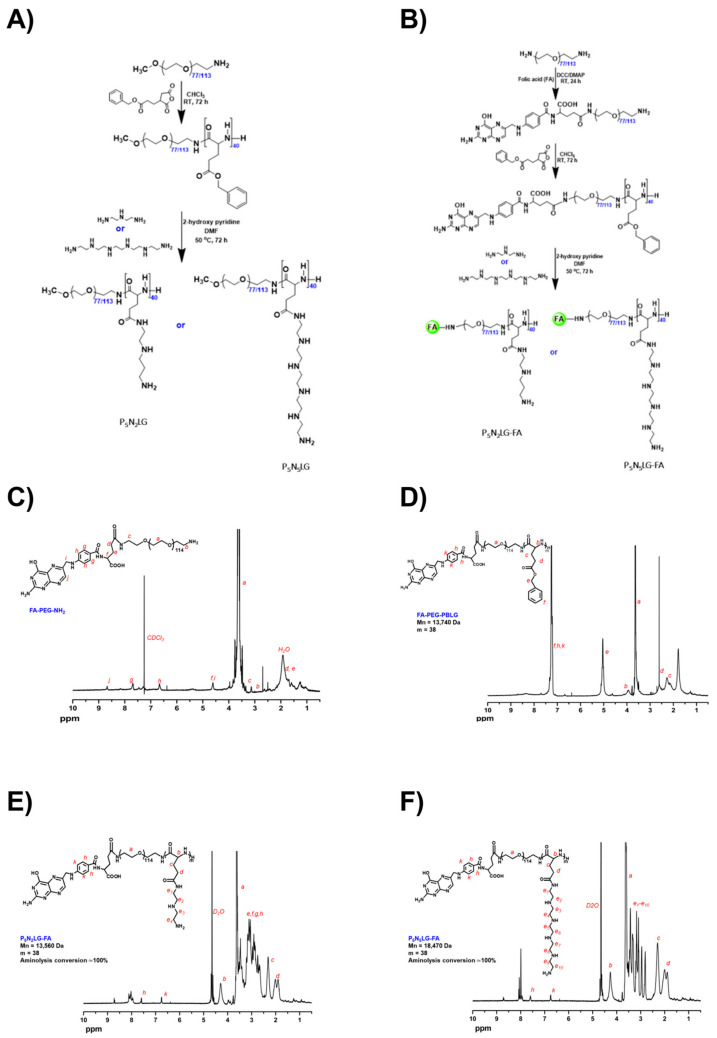
Synthesis scheme of PNLG polymers. (**A**) Synthesis of PNLG copolymers (P_5_N_2_LG and P_5_N_5_LG control polymers). (**B**) Synthesis of PNLG-FA copolymers. ^1^H NMR spectra of (**C**) FA-PEG-NH_2_ polymer, (**D**) FA-PEG-PBLG, (**E**) P_5_N_2_LG-FA, and (**F**) P_5_N_5_LG-FA polymer.

**Figure 2 cells-10-01896-f002:**
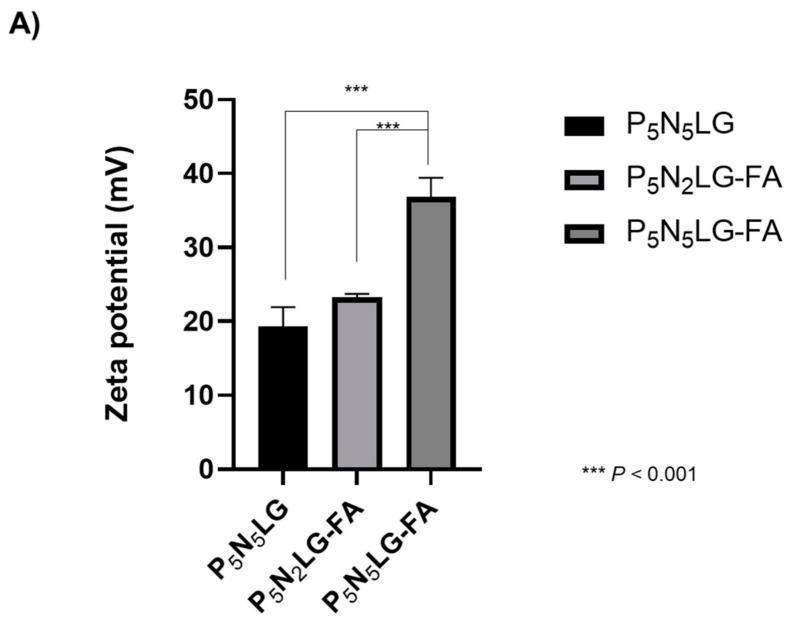
Physiochemical and cytotoxicity profile of PNLG polymers. (**A**) Net surface charges of PNLG based polymers were determined by Zetasizer. (**B**) In vitro cytotoxicity of polymers. KB, MCF7, and A549 cells were treated with of P_5_N_5_LG, P_5_N_2_LG-FA, P_5_N_5_LG-FA (1 to 50 μg/mL), along with 25 kDa branched PEI as positive control and PBS as negative control for 48 h. The cell viability was determined by MTT assay. Each cell line was tested at least three times and data shown are representative experiments performed in triplicate. Bars represent mean ± SD. ** *p* < 0.01, *** *p* < 0.001.

**Figure 3 cells-10-01896-f003:**
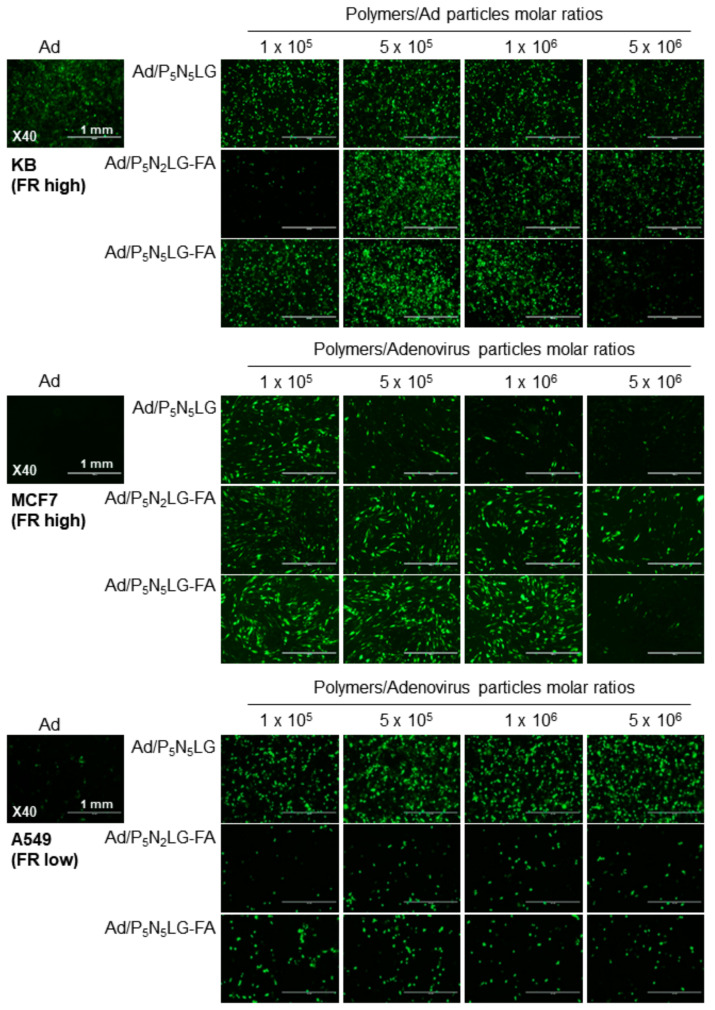
Transduction efficiency of Ad nanocomplexes. FR-high (KB and MCF7) and FR-low (A549) cells were treated with naked Ad, Ad/P_5_N_5_LG, Ad/P_5_N_2_LG-FA, or Ad/P_5_N_5_LG-FA at various Ad:polymer molar ratios (1 × 10^5^ to 5 × 10^6^). At 48 h after transduction, GFP expression levels were analyzed by fluorescence microscope. Original magnification rate: ×40. Scale bar is indicative of 1 mm.

**Figure 4 cells-10-01896-f004:**
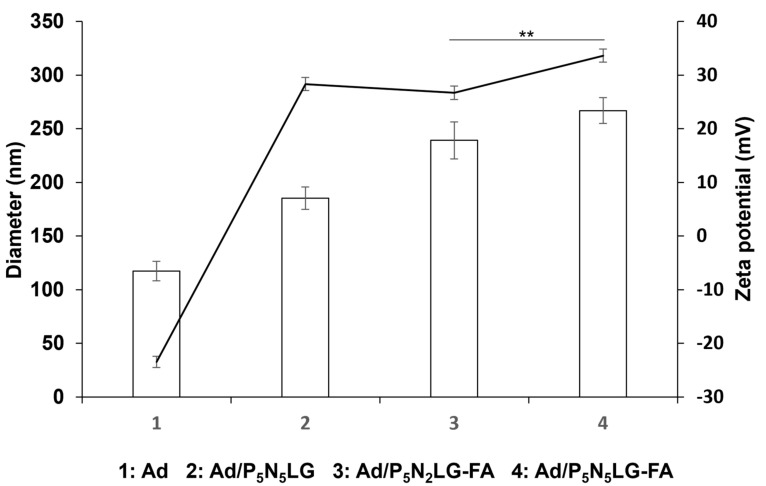
Physicochemical properties of Ad nanocomplexes. Average particle size and surface charge of naked Ad, Ad/P_5_N_5_LG, Ad/P_5_N_2_LG-FA, or Ad/P_5_N_5_LG-FA at Ad:polymer molar ratio of 5 × 10^5^ were analyzed by Zetasizer. The data are representatives of triplicate samples. Bars represent mean ± SD. ** *p* < 0.01.

**Figure 5 cells-10-01896-f005:**
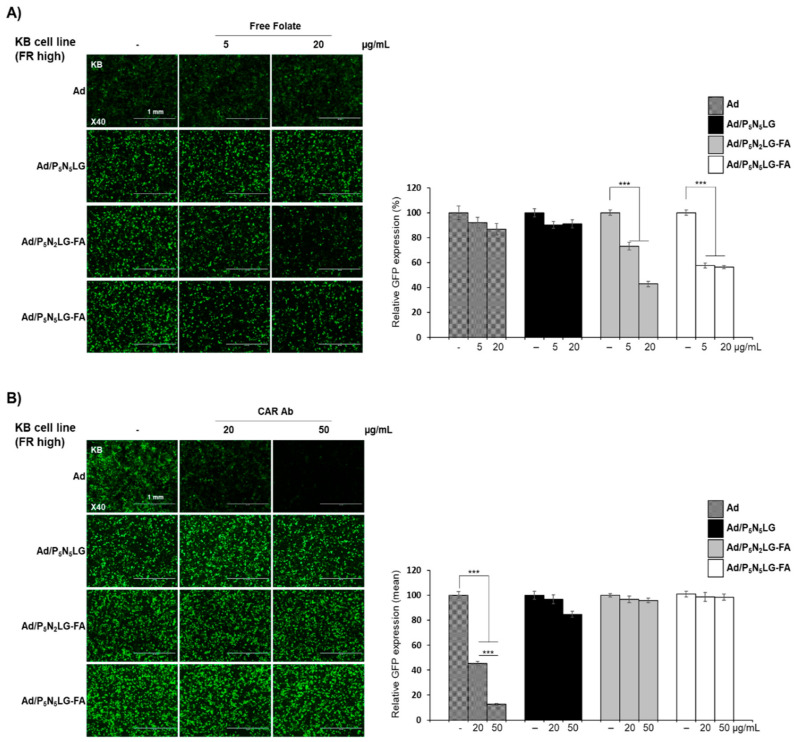
Cell uptake mechanism of Ad nanocomplexes. Competition assay with (**A**) FA or (**B**) CAR-specific Ab. KB cells were pretreated with FA or CAR-specific Ab, along with PBS as negative control, and then transduced with naked Ad, Ad/P_5_N_5_LG, Ad/P_5_N_2_LG-FA, or Ad/P_5_N_5_LG-FA. At 48 h post treatment, transduction efficiency was analyzed by fluorescence microscope and flow cytometry. The data are representatives of three independent experiments performed in triplicate. Bars represent mean ± SD. *** *p* < 0.001. (**C**) Cellular uptake efficiency of Ad and Ad nanocomplexes in KB (FR-high) and A549 (FR-low) cells. The cancer cells were treated with FITC-labeled samples (Ad-FITC, Ad-FITC/P_5_N_5_LG, Ad-FITC/P_5_N_2_LG-FA, or Ad-FITC/P_5_N_5_LG-FA) for 2 h at 37 °C, then the FITC expression level in cancer cells were analyzed by flow cytometry. The data are representatives of three independent experiments performed in triplicate. Bars represent mean ± SD. *** *p* < 0.001. Original magnification rate: ×40. Scale bar is indicative of 1 mm.

**Figure 6 cells-10-01896-f006:**
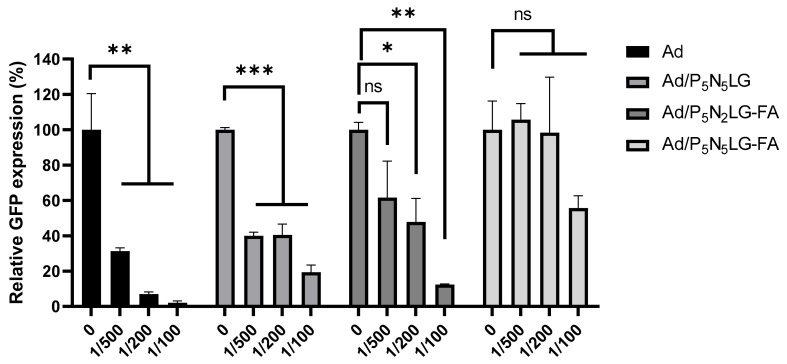
Neutralizing antibody assay in vitro. KB cells were treated with Ad or Ad nanocomplexes pre-treated with Ad-specific neutralizing Ab (100MOI) for 48 h at 37 °C. The degree of GFP expression was then observed and analyzed by Incucyte Zoom live cell analysis system (Essen Bio-Science, Ann Arbor, MI, USA). The data are representatives of duplicate samples. Bars represent mean ± SD. * *p* < 0.05, ** *p* < 0.01, *** *p* < 0.001.

**Figure 7 cells-10-01896-f007:**
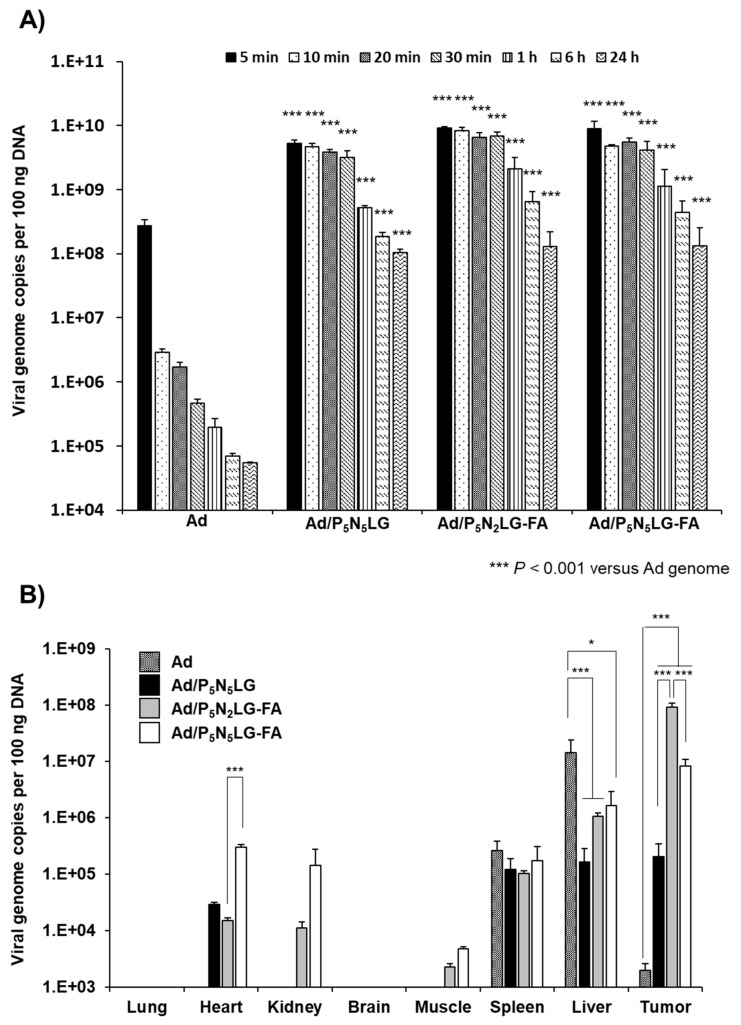
Pharmacokinetics and biodistribution profile of Ad nanocomplexes. (**A**) Pharmacokinetics of Ad nanocomplexes. Mice bearing subcutaneous KB tumors were intravenously administered once with 1 × 10^10^ VP of naked Ad, Ad/P_5_N_5_LG, Ad/P_5_N_2_LG-FA, or Ad/P_5_N_5_LG-FA (*n* = 3 per group). Blood samples were harvested from the retro-orbital plexus at 5 min, 10 min, 20 min, 30 min, 60 min, 1 h, 6 h, and 24 h after injection, then the genomic DNA were isolated. Ad genome copy number was determined by qPCR. Bars represent mean ± SD. *** *p* < 0.001 for Ad/P_5_N_5_LG, Ad/P_5_N_2_LG-FA, or Ad/P_5_N_5_LG-FA versus naked Ad at respective time points. (**B**) Biodistribution profiling of Ad nanocomplexes. Mice bearing subcutaneous KB tumors were intravenously injected three times with naked Ad, Ad/P_5_N_5_LG, Ad/P_5_N_2_LG-FA, or Ad/PN_5_LG-FA every other day (*n* = 3 per group). At 24 h after the final injection, lung, heart, kidney, brain, muscle, spleen, liver, and tumor tissues were harvested. The genomic DNA was isolated from tissue samples, then Ad genome copy number was determined by qPCR. (**C**) Immunohistochemical analysis. Heart, kidney, liver, spleen, tumor tissues were collected at 2 days after the last treatment and stained with hexon Ab. Original magnification: ×400. Bars represent mean ± SD. * *p* < 0.05, *** *p* < 0.001.

**Table 1 cells-10-01896-t001:** Molecular weight of PNLG based polymers. Molecular weights of all the polymers are calculated based on the ^1^H NMR results.

Polymer	Mw (Da)
P_5_N_5_LG	18,400
P_5_N_2_LG-FA	13,560
P_5_N_5_LG-FA	18,470

**Table 2 cells-10-01896-t002:** Tumor-to-liver ratio of systemically administered Ad nanocomplexes. The tumor-to-liver ratio, which is an important parameter determining safety and therapeutic efficacy of systemically administered adenovirus, of different treatments. The values within parentheses have been normalized to naked Ad-treated group.

Treatment Group	Tumor (VP per 100 ng of Tissue DNA)	Liver (VP per 100 ng of Tissue DNA)	Tumor-to-Liver Ratio (Normalized)
Ad	1.95 × 10^3^	1.40 × 10^7^	1.4 × 10^−4^ (1)
Ad/P_5_N_5_LG	2.00 × 10^5^	1.64 × 10^5^	1.22 (8.76 × 10^3^)
Ad/P_5_N_2_LG-FA	9.26 × 10^7^	1.05 × 10^6^	8.819 × 10^1^ (6.33 × 10^5^)
Ad/P_5_N_5_LG-FA	8.18 × 10^6^	1.64 × 10^6^	4.99 (3.59 × 10^4^)

## Data Availability

All data generated or analyzed during this work are included in this article.

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
