# Peer review of "Optimizing Active Tumor Targeting Biocompatible Polymers for Efficient Systemic Delivery of Adenovirus"

_cells, 2021, doi:10.3390/cells10081896_

Round 1

Reviewer 1 Report

Lee et al. described coating of adenovirus particles with tumour targeting biocompatible polymers enhances tumour uptake of intravenously administered Ad/polymer complexes. The authors conclude that Ad/P5N2LG-FA Ad nanocomplex had the highest tumour-targeting ability and the lowest off-target accumulation. The results are interesting, but some key experiments are missing.

My comments:

  1. There are not enough background information for why the authors chose folate receptor as their tumour-specific target. Information on the expression of FR in tissues should be included in the text.
  2. Ad/P5N2LG control is missing, why?
  3. Figure 5 is not shown completely, right side is missing.
  4. The authors claim that their Ad nanocomplexes can be used for efficient systemic delivery of adenovirus, yet they do not show any data on how these Ad nanocomplexes can avoid neutralisation by anti-Ad antibodies. This should be shown at least by performing in vitro neutralisation assay using anti-Ad antibodies or serum of Ad-infected human/mouse.
  5. Apparently the unbound/excess polymer was not removed from the Ad nanocomplex mix, why not?
  6. Both Ad nanocomplexes seemed to have increased accumulation to kidney and muscles, this should be discussed in the discussion section. 

Reviewer 2 Report

Lee et al. has reported that they evaluated the properties of three kinds of nanocomplexes which was composed of adenovirus expressing GFP coated with PNLG polymers with or without folic acid (FA) conjugation. FA-conjugated nanocomplexes (Ad/P5N2LG-FA and Ad/P5N5LG-FA) showed superior transduction efficiency compared to naked Ad or Ad complexed with PNLG polymer (Ad/P5N5LG) in folate receptor (FR)-overexpressing cancer cells. Furthermore, systemic delivery of Ad/P5N2LG-FA achieved higher intratumoral accumulation and lower non-specific liver accumulation than naked Ad. Thus, this manuscript contains some new findings worth reporting. However, there are critical points that should be addressed as follows:

Major issues

  1. Does all of the polymer bind to adenovirus when adenovirus is complexed with PNLG polymer? If not, I am wondering if free polymer influences the transduction efficiency in vitro and in vivo.
  2. Transduction efficiency of Ad and Ad nanocomplexes was evaluated at different Ad:polymer molar ratios in Fig. 3. Authors also should describe how much adenovirus particles are is used or MOI.
  3. The expression of folate receptor, adenovirus receptors CAR and integrins is quite important. Authors should show the data in KB, MCF7 and A549 cells.
  4. Why was Ad and Ad nanocomplexes added into media at different MOI (50 MOI in FA-treated groups and 100 MOI for CAR Ab-treated groups) in the competition assay (Fig. 5)? Authors should carry out the experiment at same MOI in order to evaluate the property of binding to the receptors CAR and FR.
  5. Figure 5 has been truncated.
  6. As shown in Fig. 6B, biodistribution profile of Ad nanocomplexes is quite interesting. Although authors have focused on the tumor and liver accumulation, they also should describe accumulation in other normal tissues in Results section.
  7. Systemic delivery of Ad/P5N2LG-FA achieved higher intratumoral accumulation than naked Ad and other Ad nanocomplexes. Furthermore, authors should evaluate transduction of Ad/P5N2LG-FA in tumor microenvironment a few days after injection, via detection of adenovirus protein or GFP expression by immunochemical staining. These data would be direct evidence that the tumor accumulation of Ad/P5N2LG-FA is due to FR-dependent cellular uptake in cancer cells but not stromal cells such as fibroblasts and endothelial cells.

Minor issues

  1. Which is right, **P<0.05 in Fig. 2B or **P<0.01 in the legend (line 300)?
  2. 60 min, 1h is repeated in the legend (line 436).

Reviewer 3 Report

Dear Sirs

Your story looks complete but I observed multiple flaws which prevent to publish you story as it is.

1.Apparently you do not like controls for various experiments. For instances some experiments are missing live Ad, some P2N2LG, some even both.

2. You do not have normal cells mouse and human and have not tested them in one setting with tumor cells using your P5N2LG or P5N5LG-based nanocomplexes in vitro

3. It remains unclear the selection of virus used to infect cells( moi per cell). In some experiments you  showed only ratio.

4. Some of your in vitro experiments were conducted using serum free condition which is not correlated with you in vivo setting where serum is enriched with BSA

5. You do not have primary cells in any of your settings. Known that aggressiveness of primary cells is much more higher and shown more tougher to target them with any nanocomplexes.

6.Figure 5, requires MFI

7. Figure 6. Your in vivo data requires validation using IHC staining of tumor and organ sections in the presence of adenoviral hexon antibodies

8. Figure 6..Based on your blood experiment presents of FA-conjugation does not change the circulation properties of P2N5LG- based adenoviral nanocomplexes vs. P2N55LG complex. Since folate receptors are also presented at endothelial cells of blood cells, your complexes target only endothelial cells and stuck there instead of targeting of tumor cells?To support that idea, tumors are enriched with endothelial cells and own blood vehicles.

Round 2

Reviewer 1 Report

The authors have addressed my comments. I support publication of the manuscript.

Reviewer 2 Report

The manuscript has been improved by responding to all of the concerns.

Reviewer 3 Report

all comments are being adressed